# Investigation of Limitations in the Detection of Antibody + Antigen Complexes Using the Silicon-on-Insulator Field-Effect Transistor Biosensor

**DOI:** 10.3390/s23177490

**Published:** 2023-08-29

**Authors:** Vladimir Generalov, Anastasia Cheremiskina, Alexander Glukhov, Victoria Grabezhova, Margarita Kruchinina, Alexander Safatov

**Affiliations:** 1Federal State Research Institution State Research Center of Virology and Biotechnology “Vector”, 630559 Koltsovo, Russia; cheremiskina_aa@vector.nsc.ru (A.C.); safatov@vector.nsc.ru (A.S.); 2Faculty of Automation and Computer Engineering, Novosibirsk State Technical University, 630073 Novosibirsk, Russia; 3Design Center of Bio-Microelectronic Technology Vega, 630082 Novosibirsk, Russia; gluhov@nzpp.ru (A.G.); dcbmtvega@yandex.ru (V.G.); 4Research Institute of Internal and Preventive Medicine—Branch of the Institute of Cytology and Genetics, Siberian Branch of the Russian Academy of Sciences, 630089 Novosibirsk, Russia; kruchmargo@yandex.ru

**Keywords:** biosensor, SOI-FET, transistor, detection, virus, particles, probability density

## Abstract

The SOI-FET biosensor (silicon-on-insulator field-effect transistor) for virus detection is a promising device in the fields of medicine, virology, biotechnology, and the environment. However, the applications of modern biosensors face numerous problems and require improvement. Some of these problems can be attributed to sensor design, while others can be attributed to technological limitations. The aim of this work is to conduct a theoretical investigation of the “antibody + antigen” complex (AB + AG) detection processes of a SOI-FET biosensor, which may also solve some of the aforementioned problems. Our investigation concentrates on the analysis of the probability of AB + AG complex detection and evaluation. Poisson probability density distribution was used to estimate the probability of the adsorption of the target molecules on the biosensor’s surface and, consequently, to obtain correct detection results. Many implicit and unexpected causes of error detection have been identified for AB + AG complexes using SOI-FET biosensors. We showed that accuracy and time of detection depend on the number of SOI-FET biosensors on a crystal.

## 1. Introduction

The risk of pathogens appearing in crowded places can have very serious consequences, both for the individual and for countries as a whole. Well-known pandemics in human history, such as the Spanish flu in 1918, the Asian flu in 1957, the Hong Kong flu in 1968, the swine flu in 2009, and the coronavirus pandemic (COVID-19) in 2020, have clearly shown the serious effects of viral infections on human life, along with their financial and material aspects [1]. The rapid and early diagnosis of infections with epidemic or pandemic potential allows for the timely implementation of many sanitary and anti-epidemic measures, thereby reducing the damage. Some examples of these measures include quarantine, isolation, hospitalization, vaccination, treatment, etc. [2,3,4].

Currently, there are two main rapid diagnostic approaches: molecular biology and serology. Each has advantages and disadvantages and is used to look for specific markers of viral diseases (viral particles, genetic material, and specific antibodies).

One molecular biology method is polymerase chain reaction (PCR), which is based on the selective amplification of viral DNA molecules. PCR has a high diagnostic value due to its ability to determine virus-mediated group identity. It is considered the “gold standard” of diagnosis and is widely used in virology laboratories and inpatient and outpatient settings [4].

Serological diagnostic methods are used for the detection of antibodies or antigens in the blood [5,6]. The principles of serological methods are based on modern knowledge in the field of immunochemistry and the physical–chemical laws of antibody and antigen interactions [6,7,8]. They are employed to determine the etiology and respective times of infections or recurrences, the latter comprising endogenous reactivations and exogenous reinfections. These methods have high accuracy, sensitivity, and reliability and allow us to obtain numerical estimates of antibody (AB) or antigen (AG) titers and their dynamics and development. Serology is often used to diagnose diseases without obvious clinical signs or that are asymptomatic, such as HIV infection or hepatitis [4,5,6,7]. Examples of serological methods are enzyme immunoassay, immunofluorescence analysis, immunochromatographic analysis, etc.

Biosensors based on silicon-on-insulator field-effect transistors (SOI-FET biosensors) are an alternative to traditional detection methods of infections.

The use of ion-sensitive field-effect transistors to investigate biological signals and measure ion concentrations in a solution was first proposed by Bergveld P. in 1970 and 1972 [9,10]. These studies have raised issues with the use of sensors, such as using changes in potential as a device signal and the selectivity of the biosensor signal when finding target molecules in an analyzed sample with foreign biological particles. These studies initiated the development of a SOI-FET-biosensor-based device for the detection of various biological molecules (virus antigens, enzymes, DNA or RNA, bacteria, toxins, etc.).

Currently, more advanced biosensors for detecting biological particles are being developed. The fundamental principles of these devices are the laws of physics and chemistry, using the understanding of the transfer of the analyte as a result of its diffusion and convection and controlling the kinetics of binding and chemical reactions in the sample under study. Various biosensors, along with their characteristics, sensitivity, and design elements, are presented in the literature [11,12,13,14,15,16,17,18,19,20,21,22,23]. However, the problems of particle detection probability using sensors have received little attention in the literature.

Other challenges faced by biosensors in detecting target signals when analyzing a sample are background particles in the sample, the quality of the biosensor surface preparation, and the topology and design of the SOI-FET biosensor.

It is imperative that these problems are solved by optimizing the SOI-FET biosensor and the created analytical programs to enable the highly sensitive detection of viruses or other biological molecules.

The aim of this work is the theoretical investigation of an “antibody + antigen” complex detection process using a SOI-FET biosensor.

Therefore, the entire virus detection process is divided into three main steps:

First stage: Formation of AB + AG complexes by introducing specific antibodies and the studied sample containing target viruses and background molecules to the surface of the SOI-FET biosensor. The specific reaction between AB and AG proceeds very quickly. The interaction of antibodies with background particles is not excluded, which could also lead to the formation of non-target complexes and detection errors.

Second stage: Adsorption of AB + AG complexes on the biosensor surface, which causes current modulation in the source–drain circuit. As a result, the target information signal is formed. This is defined as the difference between the value of a priori selected initial biosensor current, set before antibody introduction, and the current after the end of the detection process.

Third stage: Registration of electric current in the source–drain circuit of biosensors using specialized devices.

Adsorption of AB + AG complexes from the liquid to the nanowire surface is a random event that depends on a number of factors. These include the radius of the AB + AG complex, its diffusion coefficient and concentration, the viscosity and temperature of the suspension, etc.; see, e.g., [24,25]. For example, Duan X. et al. investigated the response of the biosensor to a decrease in the concentration of the target molecule [26]. They showed that a decrease in the concentration of the substance leads both to a decrease in device sensitivity and an increase in analysis time. For example, the biosensor response for the 2 nM sample increased sharply when the sample was added, while for the 200 fM sample, the biosensor response remained practically unchanged during the whole measurement time (2 h). Therefore, if the biosensor is discharged by electric charge carriers during adsorption, and the area of discharge is comparable to the size of the sensor element (nanowire), then a single particle can potentially completely overlap its conductivity. Thus, the potential sensitivity of one particle per sensing element is achieved [12]. Since the detection of target molecules with a concentration of 200 fM (a femtomolar solution contains about one target molecule per nanoliter [27]) results in almost no signal change, we can conclude that the molecule has not been adsorbed on the biosensor surface. Such experiments indicate the probabilistic character of particles hitting the surface of the biosensor and, consequently, the probabilistic character of their detection [27,28].

## 2. Materials and Methods

Generally, a biosensor consists of a receptor layer (antibody, aptamer, enzyme, etc.) and a field-effect transistor used as a transducer.

The SOI-FET biosensor is based on a field-effect transistor with two isolated gates [15,16,17]. The first gate is designed as a nanowire (NW) between the source and the drain of the biosensor. The second is located on the reverse side of the biosensor crystal. It is used to select the initial current in the source–drain circuit [15]. The source of the input signal is various ions, molecules, proteins, DNA, etc., which are present in the analyzed solution and partially adsorbed on the surface of the first gate—NW [11,12,13,14]. Adsorption of biological molecules on the NW surface leads to a change in the concentration of charge carried in it and, accordingly, modulation of the current, i.e., biosensor response.

The most necessary characteristics of a biosensor are high sensitivity, specificity, and speed. Sensitivity refers to the minimum amount of a substance that can be detected with its help in the test sample. Specificity is the reliable detection of the target substance (antibody or antigen) against other particles.

Figure 1 shows the topology of the biosensor made by the Vega BioMicroelectronics Technology Design Center (Russia). It contains the following elements:
1, 10—Typical biosensor contacts. The contacts are necessary to connect the silicon crystal to the contacts of the case;2—Nanowire (NW, first gate). The NW is located between the source and drain electrodes of each transistor;3, 8—Ground contacts;4—Transistor, NW region;5—Drain electrode;6—Gate contact;7—Typical point of contact;9—Crystal of biosensor. All components of the biosensor are located on the crystal surface.

**Figure 1 sensors-23-07490-f001:**
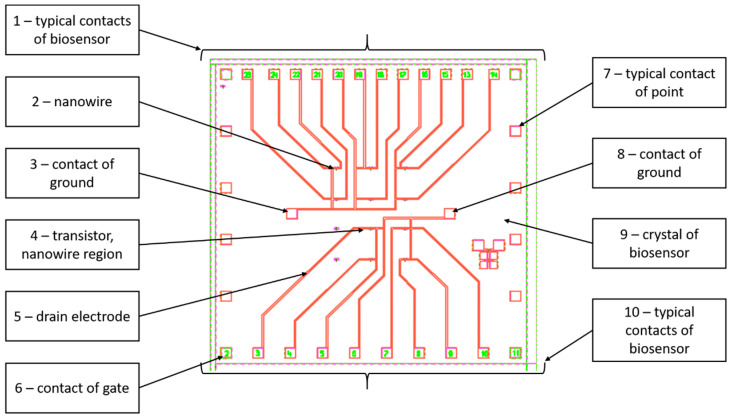
Topologies of biosensor crystal.

There are 10 independent SOI-FET biosensors located on the crystal surface. The overall dimensions of the crystal are 6 mm × 6 mm. The back side of the crystal is glued to the body chip. A photo of the biosensor case is shown in Figure 2.

The biosensor is connected to the data acquisition and processing recorder, in which the information signal is amplified in analog form, converted to digital, and subjected to primary mathematical processing. Further signal processing is performed by a computer, which makes it possible to use its capabilities for the statistical analysis of results, data transmission via communication channels to a single information center, practically unlimited storage time of results, rapid decision making, etc.

The operational principle of the recorder is based on measuring the value of current flowing in the source–drain circuit of the biosensor. The recorder software performs the following functions:-Plotting the dependence of the biosensor electric current on time;-Calculating the average value of current for a given measurement time interval. All particles (antibodies, antigen, and background particles) in the sample under study experience Brownian motion. The process of movement occurs spontaneously and proceeds constantly. The mean square displacement of particle Δ^2^ during observation *t* is found from the Einstein–Smoluchowski equation; see, e.g., [24,25]:
(1)$$\Delta^{2}=\frac{RT}{3\pi \eta \;r\;N_{a}}\cdot t$$
where: 

*R*—Universal gas constant 8.31 [J/(mol·K)];

*T*—Absolute temperature [K];

*η*—Viscosity of sample [Pa·s];

*r*—Radius of suspended particles [m];

$N_{a}$—Avogadro constant 6.02 × 10^23^ [mol^−1^];

*t*—Observation time [s].

Numerical estimates show that colloidal particles, such as a complex of flu virus and anti-influenza virus antibodies (AB + AG), with radii *r* = 5.00 × 10^−8^ m, during observation for *t* = 200 s in aqueous medium with a viscosity *η* = 10^−3^ Pa·s at a temperature of *T* = 300 K can overcome the linear distance *l*:(2)$$l=\sqrt{{∆}^{2}},m$$

After substituting the listed parameters in (1), we obtain:

*l* = 58.7 × 10^−6^, m
(3)


The value of *l* allows us to give a reasonable estimate of the measurement volume for one biosensor *v*1:(4)$$v1=\frac{4}{3}\cdot \pi \cdot l^{3}=4.23-10^{-13},m^{3}$$and for ten *v*10 biosensors located on the crystal (Figure 1 and Figure 2):

*v*10 = 4.23 × 10^−12^, m^3^
(5)


During detection, both target AB + AG complexes and background particles (which are inevitably present in any test sample) are simultaneously adsorbed on the surface of the NW biosensor. Biosensors transform the adsorption of AB + AG complexes into a useful information signal represented as an electric current (useful electric signal) and the adsorption of background particles into an electrical interference signal [26]. As a result, an additive (total) output signal *b* is observed at the recorder output, which is described by the following equation:

*b* = *Ax* + *c*
(6)

where:
*x*—Useful information signal of target complexes;*c*—Interference signal of background particles;*A*—Logical parameter that takes values 0 or 1.

In general, the solution to the problem of express detection of AB + AG is reduced to an estimate of the value of *b*. Parameter *A* takes one of two values: *A* = 1 or *A* = 0.

In the detection process, the following four logical events are probable according to the Latin square with two rows and two columns [27,28]:The target AB + AG complex is present in the sample, and it is correctly detected against the background of the interference signal. Result—the decision is made correctly, *A* = 1;The target complex is present in the sample, but it has not been detected. The result is an error in decision making, *A* = 0;The target complex is missing in the sample, but it has been “detected”. The result is an error in decision making, *A* = 0;The target complex is missing in the sample, and it is not found. Result— the decision is made correctly, *A* = 1.

Finally, the detection process should be completed with one of two mutually exclusive solutions: *A* = 1 or *A* = 0. Any other result of detection or any uncertainty in decision making is not allowed. It is obvious that the detection of a single AB + AG complex using a biosensor would not be statistically reliable. Thus, the expected potential biosensor sensitivity at the single-particle level becomes questionable. In order to obtain reliability, a series of independent samples is carried out, which will determine the probability that AB + AG complexes will appear *m* times in the measured volume of the biosensor, and the parameter A takes a value equal to one.

The Poisson distribution is the closest description of probabilistic discrete *m* time detection AB + AG complexes in the measured volume [29]. The following mandatory assumptions are made:-Measurements are independent of each other; they can be considered as random processes both in time and (or) in space;-The probability of the occurrence of event A in a single unit dimension is small (0.05–0.1 and less) and constant;-Number of measurements—the sample of n measurements is quite large;-Dispersion index χ^2^ > 1:


χ^2^ = *σ*1*/σ**
(7)

where:
*σ*1—Standard deviation of distribution under study;*σ**—Standard deviation of approximating distribution.

The Poisson probability density for AB + AG complexes is written as:(8)$${P(b)}_{AB+AG}\approx \frac{\lambda_{AB+AG}^{m}}{m!}\cdot e^{{-\lambda }_{AB+AG}}$$
where:
*λ*_AB+AG_ = *n*·*p*(1)—Average intensity of occurrence of events during observation;*m* = 0, 1, 2 ÷ 25—Number of simultaneously expected events; *n*—Samples of process measurement;*p*(1)—Probability of the detection of target event in one count.

The Poisson probability density for background particles is written similarly:(9)$$P\left(b\right)bg\approx \frac{\lambda_{bg}^{m}}{m!}\cdot e^{{-\lambda }_{bg}}$$

The likelihood ratio *Z*(*b*):(10)$$Z\left(b\right)=\frac{{P(b)}_{AB+AG}}{{P(b)}_{bg}}$$
allows us to select the most reliable hypothesis of the occurrence of event *A*.

It should be kept in mind that these probability densities are generally characterized by the parameter *λ*, which is calculated separately for AB + AG complexes and background particles. An increase in the likelihood ratio leads to an increase in the reliability of the detection of the target complex. The decision about the presence of a useful information signal is made if the likelihood ratio has a value greater than one. The choice of the hardware design of the recorder, its software, and the algorithm for analyzing and processing the input data is selected based on the four conditions described above. For example, the first condition allows us to avoid possible significant material costs for the organization of large-scale anti-epidemic measures. In this variant, the threshold level *Z*(*b*) should be chosen based on the equality of probabilities (8) and (9) with some correction factors for the risk of a false positive result of complex detection. Considering the equality of ratios allows us to avoid the analysis of a priori data on the presence or absence of a signal already at the stages of the design and construction of the circuit recorder.

## 3. Numerical Estimates

A chip with ten biosensors is designed for the detection of AB + AG complexes. It is assumed that detection should be carried out considering compliance with the mandatory conditions of Poisson distribution, for *t* = 300 s and *n* = 200 samples of process measurement. The probability *p*(1) of detecting a single AB + AG complex with one biosensor is determined using the following equation:(11)$$p\left(1\right)\approx \frac{v1}{V_{s}}\cdot S$$
where:
*v*1—Measurement volume for one SOI-FET biosensor [m^3^];*Vs*—Total volume of sample for investigation and detection on surface of the biosensor crystal [m^3^];*S*—Number of AB + AG complexes in the volume of the sample for investigation and detection via the biosensor.

Similarly, for ten biosensors [30]:(12)$$p\left(10\right)\approx \frac{d\cdot v1}{V_{s}}\cdot S$$
where: *d*—Number of biosensors on crystal surface.

The number of complexes can be calculated using the following equation:

*S* = *C* · *V_s_*
(13)

where: *C*—Number of complexes in total volume of sample for investigation in a test tube.

After appropriate substitution, *v*1 = 4.23 × 10^−13^ m^3^, *d* = 10, *C* = 10^4^ number/mL, *Vs* = 10^−2^ mL, *S* = 100 number to Equations (11) and (12), we obtained the probability of detection *p*(1) = 8.38 × 10^−2^ for a single AB + AG complex. 

One of many possible probability densities of target complexes and background particles variants is shown in (Figure 3). The variant is determined by parameters included in Equations (1)–(13):
1—Target complex was present in the test sample, and it was correctly detected, *A* = 1, *m* = 10–25;2—Target complex was present in the test sample, but it was not detected, *A* = 0; *m* = 10–25;3—Target complex was absent in the test sample, but background particles were mistakenly detected as target complex, *A* = 0, *m* = 0–10;4—Target complex was absent in the test sample, while background particles were present in the sample and correctly detected as noise, *A* = 1, *m* = 0–10.

## 4. Discussion

Background particles are always present in tested samples. The sources of particles may include the samples themselves, laboratory utensils, the chemical reagents used, the air of diagnostic laboratories, laboratory clothing, human breathing, the surface of the biosensor, etc. Numerous natural proteins are present in blood samples, which also creates a background signal (noise) [31]. The complete removal of these particles is a complex and expensive task that has no easy solution. Thus, it is obvious that the detection of viruses and antibodies in samples is always carried out in the presence of numerous and diverse background particles. They can have a negative impact on detection. Within the framework of the presented work, an attempt was made to investigate the probability of detecting influenza AB + AG complexes using a biosensor and to establish the factors affecting the reliability of detection. Studies have shown that the use of biosensors imposes strict requirements on the detection method, biosensor design, and qualifications of personnel. Currently, theories of experimental planning and methods of technical decision making are being developed at the intersection of analysis, computational mathematics, statistics, and optimization theory [29,30]. Planning provides an understanding of objectively existing limitations, provides confidence in the reliability of the obtained or expected results, makes it possible to trace the links between the various operational procedures of the experiment, etc.

The undeniable advantages of silicon-on-insulator-based biosensors are their potentially high sensitivity, high speed, low material and time costs, instrumental measurements excluding subjective evaluation of obtained results, and mathematically justified reliability of detection. A biosensor, like any other device, has advantages and disadvantages and should be handled with care. For example, it should be emphasized that the likelihood ratio is calculated for a sample already deposited on the biosensor’s surface. The likelihood ratio of another sample is likely to be different from the first one. This circumstance established the methodical construction of our experiment, the first step of which is the calculation of probability densities for background particles and “target signal + background particles”.

The development of rapid pathogen detection using SOI-FET biosensors is a step towards solving the problem of diagnostics on a new technological platform.

The use of SOI-FET biosensors will allow us to obtain qualitatively new results and fundamental knowledge:-Understanding the signs of electrical charges of AB, AG, and AB + AG;-Adjusting the sensitivity of biosensor threshold electronically;-Creation of automated, inexpensive stationary posts for rapid detection of pathogens in cities, airports, subways, stadiums, and other real-time facilities;-The use of modern digital technology for processing, storing, and transmitting display results using computers, radio channels, or the Internet;-Organization of disposable biosensor production on the level of hundreds of thousands and even millions of pieces at a minimal price;-Availability on the market of ready-made computer programs for processing indication signals and statistical processing;-Ability to create biosensors for individual use.

Investigating the probability density detection of AB + AG complexes using a biosensor and considering the relationship of Equations (1), (12) and (13) allowed us to draw the following conclusions:-The total measured volume of biosensors on the surface crystal must coincide with the volume of the test sample on it;-The probability of detecting AB + AG complex using SOI-FET biosensors increases with the increase in the number of biosensors on the crystal;-The detection of a single AB + AG complex via a biosensor cannot be considered reliable;-The ratio of the probability density of simultaneous detection of AB + AG complexes and background particles at regular intervals in independent tests should be in the range of 3–10 times—see (8)–(10);-Planning experiments, making technical decisions and conclusions based on the results detected, and the optimization of the detection process are complex tasks and remain within the remit of the human researcher.

## 5. Conclusions

This article presents the results of our study of possible errors and problems that arise due to the design features of sensors, methods of detection, sample preparations of suspensions for research, and the characteristics of the pathogens themselves. Statistical analysis revealed completely non-obvious factors that have a significant impact on the quality of sensors and their reliability.

Brownian motion and the diffusion of complexes lead to their adsorption both on the field-effect transistor (FET) gate—NW—and on the rest of the crystal surface. In the first case, the detection of the complex is possible; in the second case, detection is excluded. Ubiquitous modification of the crystal surface involving pathogen-specific antibodies will lead to the adsorption of pathogens outside of the FET gate. Therefore, not binding antibodies and antigens on the biosensor surface and the partial exclusion of the pathogen from the detection process leads to a decrease in the sensitivity of the biosensor. The sensitivity of pathogen detection via SOI-FET biosensors is decreasing overall. The number of FETs on the crystal surface must be such that the desired number of particles in the test sample have the necessary probability of adsorption onto the transistor gate.

Usually, doctors and researchers associate the desired amount of pathogen with its infection dose for humans. This can vary from single values to tens of thousands for various pathogens [32]. These values also determine the requirements for biosensor sensitivity.

These circumstances raise the question of fabricating a sufficient number of SOI-FET biosensors on a crystal to increase the probability of adsorption of the target molecules onto the NW surface. Modern technology allows the creation of thousands of biosensors on the crystal surface, but their excessive number leads to an increased cost. In addition, sequential registration of signals from a large number of SOI-FET biosensors will result in increased detection time as a result of analyzing a large amount of data. However, increasing the number of SOI-FET biosensors will increase the probability of adsorption of target molecules on their surface, thus reducing the analysis time.

Modification of the entire crystal surface with pathogen-specific antibodies may lead to the adsorption of the virus outside the transducer of the biosensor and the partial exclusion of the virus from the detection process, resulting in an undesirable decrease in the sensitivity of the detection method. Simultaneous adsorption of target molecules and background particles onto the biosensor surface leads to detection errors. To eliminate this problem, the sample preparation of the test sample and removing or reducing background particles should be performed with consideration of the diffusion coefficient of both background and target particles. We propose the use of the particle diffusion coefficient (1) as one of the criteria for the design of a biosensor based analytical device. This coefficient allows us to estimate the effective area monitored using an individual biosensor, considering its size and the radius of the pathogen particle, thus calculating the required number of biosensors on a single crystal. Thus, a line of specialized biosensor-based detection devices should be developed that are aimed at selectively detecting specific bacteria, viruses, or proteins.

Summarizing the results of this work, we note that rapid and accurate pathogen detection is a complex interdisciplinary problem that currently has no unambiguous, universally recognized solution. The SOI-FET biosensor shows a promising outlook for solving this problem and can potentially compete with traditional detection methods on the market. However, this requires solving a number of problems: designing the topology of the sensor; developing the circuit diagram of the recorder; programming the detection algorithm; developing the statistical analysis of the results and their reliability; manufacturing the receptors; and developing the procedure for preparing samples for analysis.

## Figures and Tables

**Figure 2 sensors-23-07490-f002:**
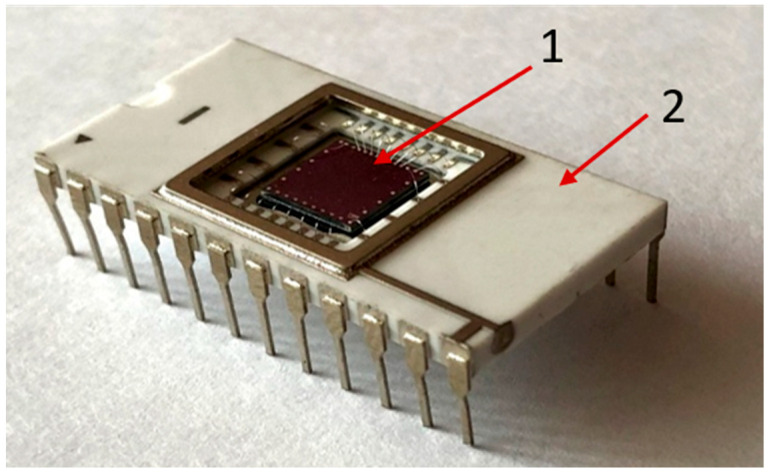
Photo of biosensor crystal in the case: 1—crystal of biosensor; 2—case.

**Figure 3 sensors-23-07490-f003:**
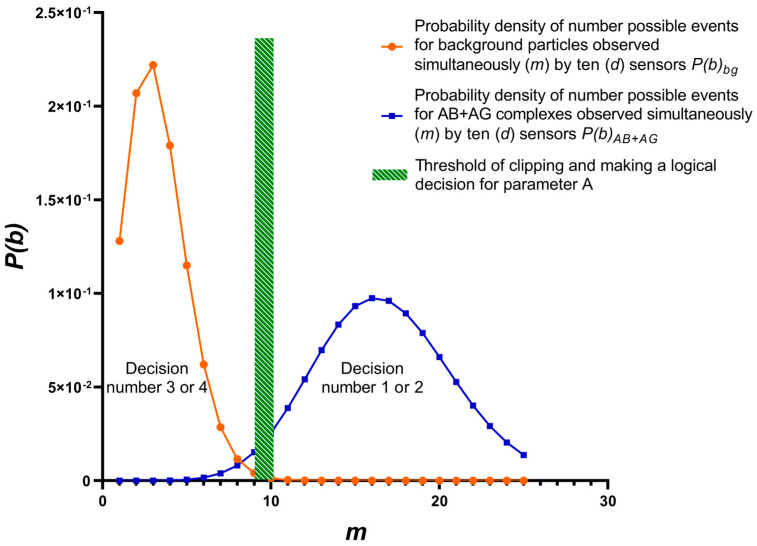
Four detection options. Probability density of simultaneous detection of particles in equal time intervals in independent tests. Dotted green area—threshold of clipping and making a logical decision for parameter *A*; blue line—AB + AG complexes; orange line—background particles. Nu-merical estimates of the probability density detection of influenza virus using a SOI-FET biosensor are presented in the additional file “Numerical estimates”.

## Data Availability

The data will be provided upon request.

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
