# Peer review of "Investigation of Limitations in the Detection of Antibody + Antigen Complexes Using the Silicon-on-Insulator Field-Effect Transistor Biosensor"

_sensors, 2023, doi:10.3390/s23177490_

Round 1

Reviewer 1 Report

The authors present a theoretical investigation of the detection of antibody-antigen complexes using a SOI-FET biosensor.

I really tried to understand what the authors are trying to achieve which the manuscript. After reading the manuscript twice in am still unable to really understand what it is about. I am sure that the analysis of the device performance is of interest but the structure of the manuscript and the language used is so difficult to follow that it is extremely hard to understand what the real focus of the manuscript is.

For example: half of the abstract is given over to a description of the four possible outcomes of a detection process. These are the well-known true positive, false positive, false negative and true negatives. It hardly seems necessary to explain these outcomes in a journal devoted to sensors?

I also found it very difficult to understand the aims of the manuscript based on the abstract.

The introduction gives a thorough background on biosensors. In fact, it reads almost like a review paper. Despite this thorough collection of citations to the literature describing how common sensors function, it is very difficult to understand the context of the present work. This is primarily due to the poor English. Below are a few examples:

1.     On example of crystal with ten biosensors, estimates of probability detection… line 19. I couldn’t understand this.

2.     Currently, the problem of rapid diagnosis is solved by various methods and technical means that perform, and often successfully, only single tasks. – line 36 This is very hard to understand.

3.     However, attention to problem of probability detection is insufficiently. - Line 73 ???

4.     Mistake detected of target pathogen by SOI-FET biosensor may be as result proceedures… - line 74 I really couldn’t understand this

5.     The authors refer to a crystal. This seems to be what would normally be referred to as the chip, or device?

The rest of the manuscript is full of these kinds of formulations that makes is very hard to read and understand.

There is a materials and methods section that describes a device. However, there do not appear to be any results from the actual device, and no comparisons between the predictions and actual data appear to be made. The complete description of the device seems to be unnecessary in the main text and could possibly be presented as supplementaru information.

I am unable, really, to judge the quality of the work beyond this since the language is so hard to follow. I unfortunately recommend that the authors rewrite the manuscript completely and  resubmit.

The English must be improved. At present it is almost impossible to follow the manuscript as is.

Author Response

Dear Reviewer. In the attachment, please find our the manuscript entitled -"Investigation of antibody+antigen complex detection by the SOI-FET biosensor".The manuscript is new variant. I attach additional files - drawings and numeral estimates also.

The authors are grateful for your valuable comments.

Dr. Generalov

Reviewer 2 Report

The work provides an overview that focuses on the theoretical investigation of detecting antibody-antigen (AB+AG) complexes using a SOI-FET biosensor. The manuscript aims to address the problem of virus detection in biotechnology, virology, and medicine using biosensors. The work also mentions the historical background of this problem and its association with the development of an ion-sensitive solid-state device for neurophysiological measurements. The authors conducted estimates of the probability of detecting AB+AG complexes of the influenza virus using ten biosensors. The detection process is described, and four possible results are outlined.

Here are a few comments to consider for improvement:

  1. Clarify the novelty: While the manuscript mentions the theoretical investigation, it would be helpful to highlight the novel aspects or contributions of this work. What sets it apart from existing studies in the field? This will make the abstract more appealing to readers.

  2. Provide context: The historical background of the problem is mentioned, it lacks specific details. Consider briefly explaining the significance of the ion-sensitive solid-state device for neurophysiological measurements and its relevance to virus detection using biosensors. This will help readers understand the context and motivation behind the study.

  3. Expand on the results: While the work mentions the four possible results of the detection process, it would be beneficial to provide a summary of the findings or conclusions. For example, were the estimates of probability detection successful in detecting AB+AG complexes of the influenza virus? Were there any significant insights or limitations discovered during the investigation? Including these details would give readers a better understanding of the manuscript's outcomes.

  4. Structure and language: The work could benefit from a clearer and more concise structure. Consider reorganizing the sentences to improve readability. Additionally, some sentences could be rephrased for clarity and flow. Pay attention to sentence structure and grammar to ensure a smooth reading experience.

Overall, with these improvements, the abstract will provide a more comprehensive and engaging summary of the manuscript, increasing its appeal to potential readers in the field of biosensors, virology, and biotechnology.

minor grammatical and formatting revisions.

Author Response

Dear Reviewer.

In attachment, please find our manuscript entitled "Investigation of antibody+antigen complexes detection by SOI-FET biosensor. The manuscript  - it is the new variant. I attach additional files - drawing and numeral estimates.

The authors are grateful for valuable the comments.

Dr. Generalov

Round 2

Reviewer 1 Report

The language is improved in the new manuscript and it is easier to understand the aims of the work based on a reading of the abstract. However, the language could still be improved further. The title is grammatically awkward. I suggest something like the following, “ Investigating limitations in the detection of antibody+antigen complexes using a SOI-FET biosensor”. 

As a native English speaker, maybe I am being overly fussy, but I find the first sentence in the abstract, for example, to be problematic. “The development of a biosensor based on silicon-on-insulator field effect transistor (SOI-FET biosensor) to detection viruses is an actual problem in medicine, virology, biotechnology or environmental monitoring.” For a start the grammar is poor. Is should be “…field effect transistor (SOI-FET biosensor) for the detection of viruses…”. Secondly, what do the authors really mean when they say that it is an “actual problem”? Do SOI-FET biosensors exist but suffer from problems and need much improvement? Or do they not exist and need to be developed? Do they work for some applications but not for others? This is very unclear. The authors specify in the subsequent sentence that some of the problems are “technological” and then that “…studying the detection process using a biosensor will solve some of the problems”. I also find this extremely vague. I have full understanding for the difficulties of writing in English when it is not ones first language and nobody expects language to be perfect, but the reader should be able to understand the content of the text, and not have to guess at vague formulations. It seems that the basic message of the abstract could be:

“SOI-FET biosensors exist but need to be improved in order to reach their full potential in the fields of …etc etc. Some improvements will be technological and others based on a better understanding of the dynamics of the system to be detected and its interactions with the sensor. We show here how improvements can be made via an understanding of the probabilities of detecting AB+AG complexes, which will aid in both the interpretation of results and in the designing of new devices. We estimate the probability of absorption using Poisson… etc etc. We show for example that the accuracy and time of detection depend on the number of SOI-FET biosensors on one crystal.”  

The rest of the manuscript suffers from the same language problems. The primary issue seems to be the misuse and/or lack of conjunctions. Maybe this is a problem that is particular to the translation from Russian to English? As I said in my first review, I strongly suggest a native speaker to go through the text. It would be a simple thing for a native speaker to add/remove/change some conjunctions and it would hugely improve the readability. With that said I will not comment further on the language but will focus on the other aspects of the manuscript.

The introduction is much improved and it is easier to understand the aims of the work presented in the manuscript.

Line 85 – “…is not excluded…” is a double negative. Would be better to say that the interaction between antibodies and background particles is included.

Line 239 – It took me a while to figure out what the (d) denoted. And I only understood when I looked at equation 12. The authors should remove this reference at line 239, or explain more clearly what it is. Probably better to remove it from line 239.

Line 327 – 329 – Do the authors mean that the reliability of testing is dependent on the competence of the person performing the test? This is a bit unclear.

Line 354 – Is the time required to analyse signal from a large number of detectors actually a real problem? If signal analysis takes a few seconds or a few minutes, does this really affect the overall time of the diagnosis in a meaningful way? Is it not probably mostly limited by the time taken to collect and prepare sample anyway?

 Overall, I found the new manuscript to be a huge improvement on the previous one. With the changes made based on suggestions from both myself and my fellow reviewer, I was able to follow the reasoning in this new version and would be happy to recommend publication. I do however strongly recommend a native speaker to go through the manuscript.

Could be improved. Very specifically, the poor use of conjunctions makes the manuscript unnecessarily hard to read, obscuring an otherwise nice piece of work.

Author Response

Thank you very much. I will work with English text. Text of articles you can see 1 august in application.

I sent the article to my friend in USA today. I think he can help me in this text.

I will ask the editorial board of the journal help translate the article to classic English as well.

Reviewer 2 Report

I would recommend the work to be published as the authors have clarified most of the concerns.

minor

Author Response

Thank you very much.

I will work with my English text. The text of article you can see 1 august in application. I will ask the editorial board of the journal to help translate the article to classic English also.